# Plasma Gel Made of Platelet-Poor Plasma: In Vitro Verification as a Carrier of Polyphosphate

**DOI:** 10.3390/biomedicines11112871

**Published:** 2023-10-24

**Authors:** Masayuki Nakamura, Hideo Masuki, Hideo Kawabata, Taisuke Watanabe, Takao Watanabe, Tetsuhiro Tsujino, Kazushige Isobe, Yutaka Kitamura, Carlos Fernando Mourão, Tomoyuki Kawase

**Affiliations:** 1Tokyo Plastic Dental Society, Tokyo 114-0002, Japan; maoh4618@me.com (M.N.); hideomasuki@elm-dc.com (H.M.); hidei@b-star.jp (H.K.); watatai@mui.biglobe.ne.jp (T.W.); keiseikai@kosesika.or.jp (T.W.); tetsudds@gmail.com (T.T.); kaz-iso@tc4.so-net.ne.jp (K.I.); shinshu-osic@mbn.nifty.com (Y.K.); 2Department of Periodontology, Tufts University School of Dental Medicine, Boston, MA 02111, USA; carlos.mourao@tufts.edu; 3Division of Oral Bioengineering, Graduate School of Medical and Dental Sciences, Niigata University, Niigata 951-8514, Japan

**Keywords:** plasma gel, polyphosphate, thermal preparation, denaturation, thrombin, controlled release

## Abstract

Plasma gel (PG) is a blood-derived biomaterial that can be prepared by heating or chemical cross-linking without the aid of intrinsic coagulation activity and has gradually been applied in the field of esthetic surgery. To explore the applicability of PG in regenerative therapy or tissue engineering, in this study, we focused on the advantages of the heating method and verified the retention capacity of the resulting PG for polyphosphate (polyP), a polyanion that contributes to hemostasis and bone regeneration. Pooled platelet-poor plasma (PPP) was prepared from four healthy male adult donors, mixed with synthetic polyP, and heated at 75 °C for 10 or 30 min to prepare PG in microtubes. The PG was incubated in PBS at 37 °C, and polyP levels in the extra-matrix PBS were determined by the fluorometric method every 24 h. The microstructure of PG was examined using scanning electron microscopy. In the small PG matrices, almost all of the added polyP (~100%) was released within the initial 24 h. In contrast, in the large PG matrices, approximately 50% of the polyP was released within the initial 24 h and thereafter gradually released over time. Owing to its simple chemical structure, linear polyP cannot be theoretically retained in the gel matrices used in this study. However, these findings suggest that thermally prepared PG matrices can be applied as carriers of polyP in tissue engineering and regenerative medicine.

## 1. Introduction

Blood concentrates, such as leukocyte platelet-rich fibrin (L-PRF), have garnered significant attention in the field of regenerative medicine, largely because of their role as potent reservoirs of growth factors that are pivotal for cell proliferation, maintenance, and differentiation [1]. When released, these growth factors initiate a cascade of events that culminate in tissue regeneration [2]. Additionally, cytokines present in L-PRF have been identified as key participants in organizing the complex tissue regeneration process. Given its rich content of bioactive molecules, L-PRF has been proposed as a promising vehicle for drug delivery [3]. However, a notable concern that impedes its broad-spectrum utilization is the stability of L-PRF membranes once introduced into the human body [1,4]. Addressing this challenge could pave the way for harnessing the full therapeutic potential of L-PRF in myriad clinical scenarios.

On the other hand, plasma gel (PG), also called albumin gel, is a blood-derived biomaterial prepared without utilizing the intrinsic coagulation pathway [5]. To date, PG has primarily been applied in the field of aesthetic surgery [6,7]. Several methods have been proposed for PG preparation, e.g., cross-linkers, but from a practical point of view, the heating method would be highly evaluated as the most convenient and modifiable method. Stiffness and degradability can be controlled by the temperature and the duration of heating. Even though most peptide-based bioactive factors contained in PG are inactivated by heating, some bioactive factors are expected to maintain and express their activity through controlled release systems. Therefore, PG matrices are attractive not only for regenerative medicine but also for cancer treatment as drug delivery systems [8]. In addition, the PG skeleton can still be used as a scaffolding material or as a barrier membrane for regenerative therapy.

Polyphosphate (polyP) consists of a linear arrangement of inorganic phosphates; therefore, it carries multiple negative charges that easily bind to cations or cationic macromolecules, such as Ca^2+^ or polyethylenimine, to form insoluble precipitates [9,10]. In humans, high levels of polyP are stored in platelets and released upon activation [11]. The released polyP is thought to function as a multiple bioactive factor, for example, a hemostatic agent, and as a material for bone regeneration. However, it was reported that the half-life of polyP in the human body is short (~1.5 h) [11]. Thus, a controlled release system for polyP without the formation of precipitates is required to retain it at the site of injection.

In this study, we focused on the characteristics of PG and hypothesized its applicability as a polyP carrier in regenerative therapy. In a previous study [4,12], the albumin–platelet-rich fibrin (Alb-PRF) complex, which consists of denatured plasma gel combined with liquid platelet-rich fibrin, appeared to retain and release growth factors like L-PRF. To examine the retention capacity of the plasma gel, in the present study, we optimized the protocol for gel preparation and evaluated this capacity for polyP.

## 2. Materials and Methods

### 2.1. Blood Collection

The Ethics Committee for Human Participants at Niigata University (Niigata, Japan) approved the study design and consent forms for all procedures (project identification code: 2019-0423) in compliance with the Helsinki Declaration of 1964, as revised in 2013. Every subject involved in the study provided informed consent.

Peripheral blood samples were collected from four healthy male volunteers aged 53–73 years. Blood was collected using glass vacuum blood collection tubes (Vacutainer^®^; BD Biosciences, Franklin Lakes, NJ, USA) containing a 1.5 mL A-formulation of acid–citrate–dextrose solution (ACD-A). Only non-smoking healthy male participants were included in the study. They provided written informed consent but were not given continuous medical treatment. Participants with smoking habits, severe lifestyle-related diseases, blood diseases, or those taking anticoagulant or antiplatelet drugs were excluded. Donors who were positive for HIV, HBV, HCV, or syphilis were excluded after pre-checking their medical history.

Finally, a hematology analyzer (pocHi V-diff, Sysmex Corporation, Kobe, Japan) was used to automatically measure the blood cell counts.

### 2.2. Platelet-Poor Plasma Preparation

The blood samples were stored or transported to the laboratory and subjected to platelet-poor plasma (PPP) preparation. The samples were centrifuged horizontally at 415× *g* for 10 min (Kubota, Tokyo, Japan). The upper plasma fraction, which was just beyond the interface of the plasma and red blood cell fractions, was transferred into 2 mL sample tubes and centrifuged at 664× *g* for four minutes using an angle-type centrifuge (Sigma Laborzentrifugen, Osterode am Harz, Germany) to collect the supernatant. The PPP preparations obtained from four male adults were mixed, frozen, and used as a pooled PPP in the following experiments.

### 2.3. Heating Protocol

Pooled PPP samples (100 or 300 μL) in 2-mL plastic microtubes were mixed with synthetic polyP (C-60, Bioenex, Hiroshima, Japan) at the final concentration (500 μg/mL), heated in a heating block at 65, 75, or 85 °C for 5–30 min, and cooled at room temperature (24–28 °C) for 10 min. Then, PPP gels were subjected to the subsequent polyP-releasing test. The basic design of the experimental procedure is summarized in the illustration of Figure 1. Although both gel matrices had a similar surface area (interface between gel and PBS), the volume of the large gels (300 μL) was approximately three times as large as the small gel (100 μL).

For comparison, polyP-containing PPP gels were also prepared by mixing 100 μL liquid PPP with 10 μL bovine thrombin (1000 units/mL) (Mochida Pharmaceutical Co., Ltd., Tokyo, Japan). After 20 min of incubation at room temperature (24–28 °C), PPP gels were subjected to the polyP-releasing test.

### 2.4. polyP-Releasing Test

To PPP gels, 1.5 mL PBS containing 5 mM EDTA, which included water contents (30.0–180.0 μL) of PPP gels, was added and incubated at 37 °C with intermittent agitation for up to 168 h. As depicted in Figure 1, 10 or 20 μL samples were collected from the PBS of extra-PPP matrices and diluted with pure water at appropriate levels. PolyP levels were quantified using 4′,6-Diamidine-2-phenylindole (Dojindo, Kumamoto, Japan) and a fluorometer (FC-1; Tokai Optical Co., Ltd., Okazaki, Japan) at excitation and emission wavelengths of 425 and 525 nm, respectively [13,14].

To facilitate the degradation of the PG matrices, 2.5% trypsin solution (FUJIFILM Wako Chemicals, Osaka, Japan) was added to the PBS to give the final concentration at 0.0125%.

### 2.5. Determination of Water Content in PPP Gels

Liquid PPP samples were weighed before the preparation of PPP gels. After heating or mixing with thrombin, PPP gels were intermittently air-dried in a heating block at 100 °C until the gels turned clouded and partially detached from the inner wall of micro tubes (~30 min). After cooling and further drying in a desiccator for 60 min, the PPP gels were again weighed.

### 2.6. Scanning Electron Microscopic Examination of Plasma Gel

For a detailed examination of the morphological and ultrastructural attributes of PG matrices, the following protocol was used: initially, the individual PG matrices underwent a meticulous cleaning procedure to remove any residual impurities. Following washing, the samples were fixed using a 2.5% neutralized glutaraldehyde solution, which is instrumental in enhancing both the biomechanical properties and stability of the collagen matrix within the PG matrix [15]. This fixation step preserves the native ultrastructural properties of the PG matrices, facilitating detailed microscopic analysis [16].

Post-fixation, a graduated dehydration protocol was implemented, wherein the samples were sequentially incubated in ethanol solutions of increasing concentrations (ranging from 50% to 100%), followed by t-butanol treatment. This protocol ensures optimal preservation of the matrix structure and prevents artifacts or distortions during the subsequent freeze-drying process.

Following dehydration, the samples were subjected to freeze-drying, a technique employed to effectively remove any residual moisture while retaining the structural integrity of the PG matrices.

For microscopic examination, the prepared samples were sputter-coated with a thin layer of gold to enhance electrical conductivity and subsequently examined under a high-resolution scanning electron microscope (TM-1000; Hitachi, Tokyo, Japan) operating at an acceleration voltage of 15 kV. This step facilitates the acquisition of high-fidelity images, providing insights into the intricate structures and potential topographical modifications occurring in the PG matrices.

### 2.7. Statistical Analysis

The data are expressed as means ± SD of four repeated measurements. To compare the mean values of neighbors’ data on the same curves, non-parametric analysis was performed using Repeated Measures Analysis of Variance on Ranks, followed by Dunn’s multiple comparison test. The Mann–Whitney U test was applied to compare the data at the same time points. Differences were considered statistically significant at *p* < 0.05.

Biomechanical properties and collagen matrix stability can be enhanced by means of physical/chemical crosslinking, by ultraviolet (UV) radiation, genipin (Gp), glutaraldehyde, and 1-ethyl-3-(3-dimethylaminopropyl) carbodiimide hydrochloride (EDC), among others. 

## 3. Results

In the present study, we confirmed that heating at 65 °C could not form PG and that at least 70 °C was required for reproducible PG formation. Regarding the duration of heating, gel formation was observed to start at approximately five minutes of heating; however, PG prepared under these conditions dissolved quickly in PBS. Thus, we confirmed that a minimum of 10 min is required for stable PG formation. Based on these data, we adopted a heating temperature of 75 °C for 10 min or longer in the following experiments.

The water content in the individual PG and fibrin gel matrices is shown in Table 1. In the PG matrices prepared from 300 μL PPP by 10 min of heating, the water content was approximately 52.8 μL (17.6% of the initial PPP volume; n = 5). The water content decreased with the heating time (30 min): approximately 29.4 μL (9.8% of initial PPP volume; n = 5). For reference, fibrin gel (FG) prepared using thrombin was weighed. The water content was approximately 177.9 μL (53.9% of the initial PPP plus thrombin volume; n = 5).

The microstructures of the surface and cross-sectional areas of the PG matrices are shown in Figure 2. In the PG matrices prepared by 10 min of heating, small pores in the submicron order were observed on the surface, and the density of the fiber-like structures was relatively lower in the cross-sectional area. In contrast, the surface pores disappeared, and the fiber density increased when the heating time was prolonged to 30 min.

The appearance of the PG matrices incubated in PBS is shown in Figure 3. Substantial degradation was not observed in either small (100 μL) or large (300 μL) PG matrices in the absence of 0.0125% trypsin (Figure 3a,c). However, the addition of trypsin almost completely broke the small PG matrices into small pieces at 144 h (six days) of incubation. In large PG matrices, only the surface region was partially degraded by trypsin. Roughly 80% of the PG matrices maintained their structure at 144 h of incubation.

The time course of changes in polyP levels in extra-matrix PBS is shown in Figure 4. This experiment was performed based on the preliminary data that DAPI-based signals can be maintained at similar levels after month-order preservation at 4 °C and 30-min heating at 75 °C, indicating the high chemical stability of polyP and its resistance to complete hydrolysis. In the PG matrices prepared by 10 min of heating, trypsin appeared to facilitate polyP release in the small PG matrices. However, statistically significant differences were not observed in either the small or large PG matrices at any time point. Similar results were obtained for the PG matrices prepared by 30 min of heating (Figure 4a). In the small PG matrices, statistically significant differences were obtained in the small PG matrices between 48 and 96 h (Figure 4b). When comparing the two different heating periods, regardless of volume, no significant differences were observed (Figure 4a vs. Figure 4b).

For reference, we also tested thrombin-induced gel matrices (Figure 4c). Unlike the PG matrices, peaks in extra-matrix polyP levels appeared between 48 and 72 h and between 72 and 120 h for the small and large fibrin gel matrices, respectively. In addition, no additional effects of trypsin were observed at any time point or size of the fibrin gel matrices. For comparison, the data obtained from the large gel matrices are shown in Figure 4d. Although all three gel matrices contained similar amounts of insoluble material used for polyP retention, there were no significant differences despite the higher water content of the thrombin gel matrix.

## 4. Discussion

This in vitro simulation test demonstrated that the large PG matrices still retained 25–30% of their initial polyP levels at seven days of incubation. This finding suggests that the PG matrix has a carrier-like capacity for retaining and releasing polyP. However, because this capacity depends on the volume, specifically the surface area-to-volume ratio, the PG matrix should be as large as possible within the limit of the implantation space to be a long-lasting material.

### 4.1. Heating Temperature and Time and Insoluble Components

It was necessary to increase the heating temperature to 70 °C or higher for PG formation. However, when the temperature exceeded 85 °C, the PG matrices quickly turned into insoluble, fragmented forms, which were distinguished from the gel forms. Therefore, a temperature range of 70–80 °C is practical for preparing PG at point-of-care.

PG is also designated as an albumin gel. However, in our preliminary study, similar gel forms could not be prepared from serum, even though the heating time was prolonged to 60 min. Thus, it is questionable whether albumin is a major component of the insoluble matrix. Based on its fibrous appearance, this component appeared to be fibrin. In addition, the polyP-releasing data support this possibility because the thrombin-prepared gel matrix retained similar amounts of polyP during the initial half of the incubation time.

In the literature, thermally denatured fibrinogen has been reported as a promising biomedical matrix [17,18]. Thus, in addition to its function as a carrier of bioactive factors, the PG matrix could function as a scaffolding material for tissue regeneration.

To clarify the molecular contents of the PG matrix, further chemical analysis should be conducted, which has been given up for PRP.

### 4.2. Structural Characteristics and Water Content

SEM examination demonstrated that the PG matrices were composed mainly of fibrous components, seemingly fibrin fibers, such as fibrin clots, produced by the late coagulation cascade. Owing to its chemical properties, fibrin has been utilized for the preparation of hydrogels in combination with thrombin. In this fibrin hydrogel, as implied by the name, water is the main constituent and exists not only in the space between fibers but also within the individual fibers that constitute the framework of the gel [19].

The water content data for the individual PG matrices (Table 1) support these structural characteristics. The difference in fiber density between the PG matrices prepared by 10-min heating and those prepared by 30-min heating is probably due to water evaporation and subsequent shrinkage. The heating temperature was 75 °C, which was lower than the boiling point; however, we observed that this temperature was hot enough to cause dew condensation on the side wall of the microtubes.

Compared to the PG matrices (78–79 vol%), the thrombin gel matrices contained a greater total amount of insoluble components (approximately 46 vol%). This indicates that the water contents in the PG matrices and thrombin gel matrices were approximately 21–22 vol% and 54 vol%, respectively. This is thought to be simply due to the lack of heating during the preparation of the thrombin gel matrices. Despite this substantial difference, it remains to be elucidated why the polyP retention and release capacities were very similar between the two types of gel matrices.

### 4.3. Degradation of the PG Matrices

Based on the difference in the degrees of the slopes, the initial phase of polyP release can be distinguished from the second phase. Throughout the incubation period, it is likely that the hydrolytic degradation of fibrin and other major insoluble proteins contributes to polyP release. However, other mechanisms may be involved in the initial steep slope. We speculate that polyP trapped (not chemically bound) in the semi-dried fibrin mesh may be released by hydration without (or before) the hydrolysis of fibrin fibers and polyP by itself.

Regarding degradation of the carrier matrix, in general, the degradation of a material depends on the surface area to volume ratio (SA:V); when the volume is lower, SA:V increases exponentially [20] and degradability (i.e., solubility) increases. In this study, the surface areas were fixed at almost the same level by sealing the gap between the PG cylinder**’**s sidewall and the microtube’s inner wall. Thus, it enabled an increase in the volume without significantly increasing the surface area.

In the absence of trypsin, insoluble components are degraded solely by hydrolysis. Such hydrolytic degradation was not detected macroscopically. However, it cannot be ruled out that the PG matrix was degraded at submicron levels to allow polyP to leak. In support of this possibility, the addition of trypsin degraded the PG matrix, as evidenced by the absence of a significant increase in the polyP release. This study provides basic information about the release of polyP from the PG matrix.

Regardless of their status (i.e., native or denatured), proteinases are responsible for protein degradation in the human body. Additionally, fibrin is specifically degraded by plasmin [21]. However, this process is complexly regulated by many factors and is influenced by fibrin structure and clot composition [21,22,23]. Briefly, a key initial reaction is the binding of plasminogen activators to the fibrin surface. Plasminogen activators convert plasminogen to plasmin, initiating fiber lysis. This thrombotic process, however, is impeded by lytic inhibitors, such as plasminogen activator inhibitors, α2-anti-plasmin, and thrombin-activatable fibrinolysis inhibitors [21,23]. In a previous study using an in vitro experimental system [24], we confirmed that plasmin could not efficiently digest the fibrin matrix as anticipated, requiring the aid of tissue plasminogen activator to facilitate the process smoothly.

In the PG matrices, the fibrin fiber diameter and fibrin network architecture appeared thicker and finer, respectively. However, these characteristics are probably due to the adhesion of protein debris and evaporation of water content. Additionally, proteins are denatured in the PG matrices. Thus, it is questionable whether the general concept of fibrinolysis can be directly applied to this material to predict the fate of the implanted material. To the best of our knowledge, there are no published data available to comprehensively understand the mechanism underlying the degradation of denatured proteins at the implantation site; however, according to a study performed by Gheno et al. [4], the Alb-PRF complex was more stable than L-PRF at the implantation site in animal experiments using nude mice. This finding suggests that denatured fibrinogen may acquire resistance to plasmic degradation or other proteolytic degradation processes.

### 4.4. Advantages of the Thermally-Prepared PG Matrices over Thrombin-Prepared Methods

In a preliminary study, since thrombin-prepared PG matrices contained a larger amount of water, we hypothesized that polyP could be released faster from thrombin-prepared PG matrices than from thermally prepared PG matrices. However, no significant differences were observed between the two matrix types in the present study. To address this, more in-depth biochemical investigations should be performed. At present, we speculate that polyP can be adsorbed onto fibrin fibers via polyP-binding proteins, as discussed in the subsection below, retained, and released as the matrix degrades.

However, owing to its several advantages, we suggest that thermally prepared PG matrices are more useful in clinical settings. This is because the thermal method can modify the stiffness of the injectable to a solid type, can be molded easily, can modulate water content, and requires no xeno-factors, such as bovine thrombin. The crude thrombin mixture can be prepared from autologous blood samples; however, it requires additional blood collection. In contrast, although the thermal method requires a heating instrument, it is less costly and compact enough for a limited space in clinics.

Many other crosslinkers, such as factor XIIIa [25], glutaraldehyde [26], and genipin [27], have been examined. However, factor XIIIa is not derived from the patient’s own blood and is costly, whereas the latter two chemicals require thorough washing to eliminate free chemicals to avoid undesired complications before use. Thus, regarding safety and hazards, the PG preparation methods using these cross-linkers are inferior to the thermal method.

### 4.5. Immunogenicity of the PG Matrix

Degradation of the PG matrix is not solely due to proteolytic degradation but also due to T-cell-dependent degradation in vivo. Immunogenicity is a more serious issue in the clinical application of denatured biologics [28,29]. The immunogenicity of denatured fibrinogen was not discussed in previous studies using those materials [17,18]. In general, heat denaturation is thought to modify the number of immunogenic epitopes depending on the individual proteins [30,31,32]. As a result, certain proteins can raise their antigen levels while others can decrease them. Thus, T-cell-dependent degradation of the PG matrix cannot be predicted using current knowledge. The stability of the albumin–PRF complex was evaluated in immunodeficient animals [4], which do not control lymphocyte infiltration or subsequent debridement. However, considering their frequent clinical application in facial esthetic treatment [6,7] and no published reports on complications, denatured plasma gels may not act immunologically like invading pathogens in the human body. Additional testing conducted in appropriate settings may be required to confirm the credibility of this concept.

### 4.6. Candidate polyP-Binding Proteins and Other Molecules

Our study did not obtain any data suggesting the entrapment of polyP in PG matrices. However, it would be interesting for future progress to discuss the possible mechanisms of polyP retention. Based on experiments using bacterial long-chain polyPs, several studies have provided data related to polyP-binding proteins. Screening using a human proteomic microarray indicated that both disabled-1 (DAB1) and phosphatidylinositol-5-phosphate 4-kinase 2 B (PIP4K2B) have the highest affinity for polyP [33]. However, these proteins are not present in the human serum [33]. Instead, myristoylated alanine-rich C-kinase substrate-like 1 (MARCKSL1), pleckstrin (PLEK), and nucleoside diphosphate-linked moiety X-type motif hydrolase-3 (NUDT3) showed detectable affinities for polyP in human serum [33]. Regarding the manner of binding, a Per-Arnt-Sim kinase domain functions as a major binding site for the covalent attachment of polyP [33]. In addition, electrostatic interactions can be observed between positively charged proteins lacking a PASK domain, such as DAB1, and negatively charged polyP [33].

Probably by electrostatic binding, parvulin, factor H, complement C1q, complement C1q tumor necrosis factor-related protein, C4 binding protein, and apolipoprotein E have been identified as candidates for polyP-binding proteins in plasma [34]. In contrast, platelet factor 4, multimerin, thrombospondin, and cyclophilin are considered to have a high affinity for polyP in the platelet releasate [34].

However, polyP binding to proteins may not be limited in a specific or electrostatic manner. Gray et al. [35] reported the other possibility that polyP has protein-like chaperone qualities and binds to unfolding proteins with high affinity in an ATP-independent manner. PolyP binds to unfolded proteins that are damaged by the antimicrobial oxidant HOCl to form a polyP–protein complex to rescue damaged proteins [35]. As this chaperone cycle works mainly in the endoplasmic reticulum, such complex formation may rarely occur in the plasma.

Limited to human plasma, thrombin, factor XI, factor XIa, kallikrien, and histones could be considered major candidates for polyP-binding proteins [34]. More importantly, in relation to this study, platelet-bound fibrin acts as a reservoir for plasminogen, FXII(a), and polyP [36].

### 4.7. Technical Limitations and Suggestions for Future Research

The quantification of polyP in biological samples, especially those derived from human tissues, has been discussed for a long time. It is believed that purification and complete degradation are required for accurate quantification of inorganic phosphate monomers. However, this method can be time-consuming, labor-intensive, and requires certain skills, which makes it unsuitable for tasks such as screening large numbers of samples. Thus, recently, the DAPI-based method, which is simple and convenient for many samples, has gradually become popular after repeated minor modifications and validations. However, this method has several weaknesses. It cannot distinguish the length of the polymer polyP or detect shorter lengths of the polyP (~15-Pi) [37]. Considering these limitations, the data should be interpreted accordingly.

Regarding trypsin, we adopted this proteinase to efficiently degrade the PG matrix. However, the effect of trypsin treatment was negligible. This may be due to the presence of serum components that can reduce enzymatic potential. Thus, we should keep in mind that trypsin may not be the best choice to simulate in vivo conditions, an experimental model in this study, although this enzyme is convenient and useful for the efficient degradation of protein-based matrices. Alternatively, it may depend on the balance between the amount of added polyP and that of its binding proteins or non-protein molecules.

By conducting additional experiments, we can gain a more comprehensive understanding of thermal and thrombin methods. Investigating the structural stability, biocompatibility, and polyP release profiles of these gels under various conditions could offer invaluable insights for future research.

## 5. Conclusions

The plasma gel appeared to retain and release polyP with increasing incubation time in a volume-dependent manner. Thus, plasma gels can be used as carriers of polyP in tissue engineering and regenerative medicine. Although the chemical structure of linear polyP is uncomplicated, it presents a challenge when attempting to retain it in the gel matrices used in this study. However, it has been observed that polyP can be effectively trapped within the plasma gel if it contains molecules such as divalent cationic polymers. This is because linear polyP has an affinity for divalent cations, resulting in the formation of insoluble compounds.

## Figures and Tables

**Figure 1 biomedicines-11-02871-f001:**
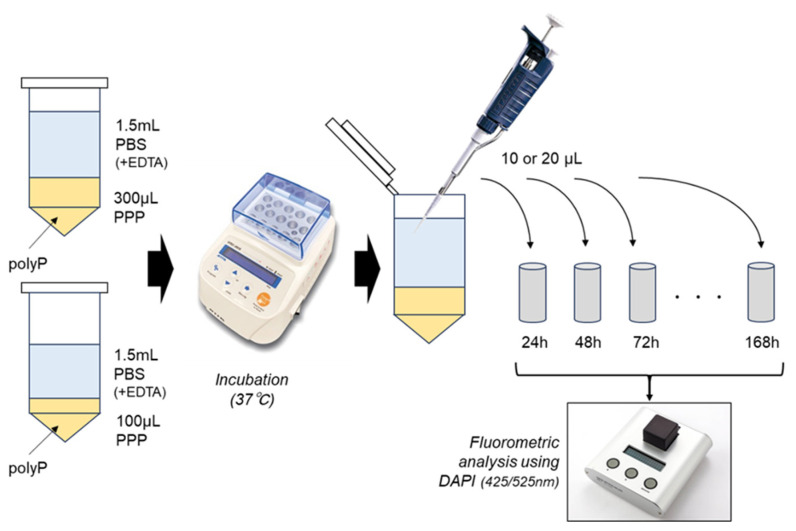
The basic design of the experimental procedure.

**Figure 2 biomedicines-11-02871-f002:**
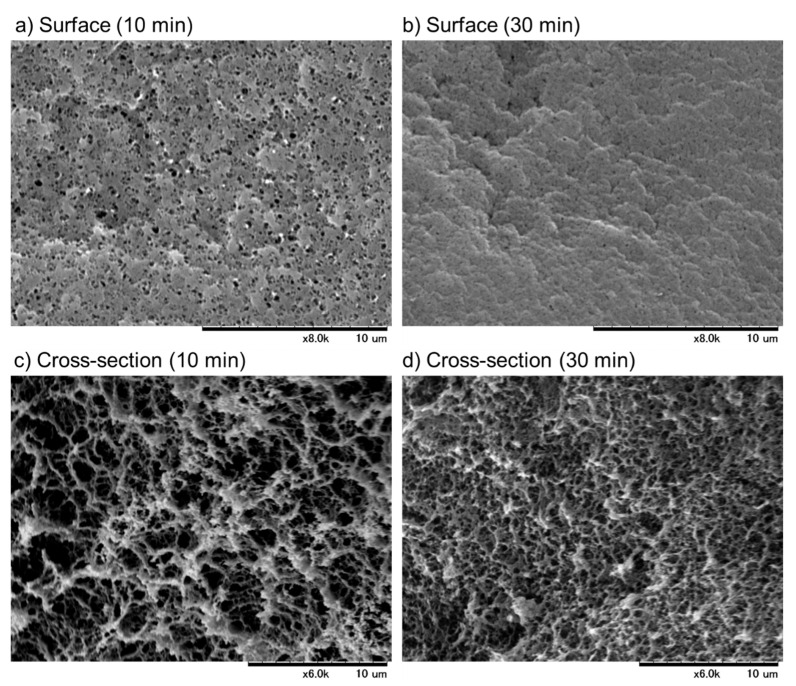
The microstructures of the surface (**a**,**b**) and cross-sectional areas (**c**,**d**) of the PG matrices. Each PG matrix was fixed, dehydrated, and examined using scanning electron microscopy. Similar findings were obtained from the other three independent experiments. Bar = 10 μm.

**Figure 3 biomedicines-11-02871-f003:**
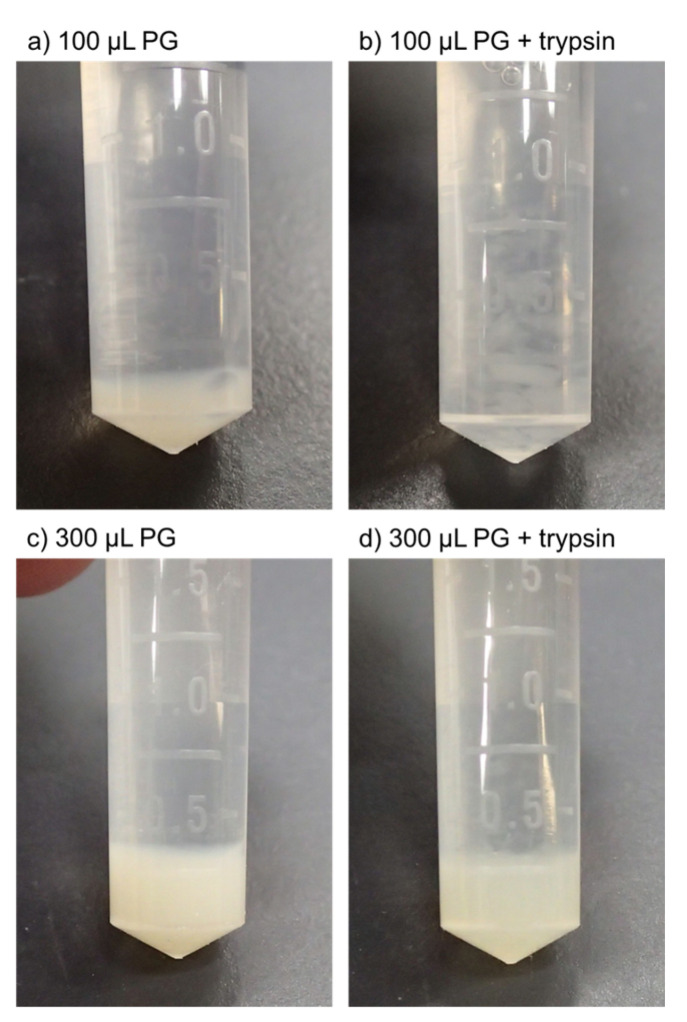
The hydrolytic and enzymatic degradation of PG matrices in PBS. (**a**) One hundred microliters of PG without trypsin, (**b**) 100 μL PG with trypsin, (**c**) 300 μL PG without trypsin, (**d**) 300 μL PG with trypsin. All the PG matrices were incubated for 144 h at 37 °C and subjected to photographing without fixation.

**Figure 4 biomedicines-11-02871-f004:**
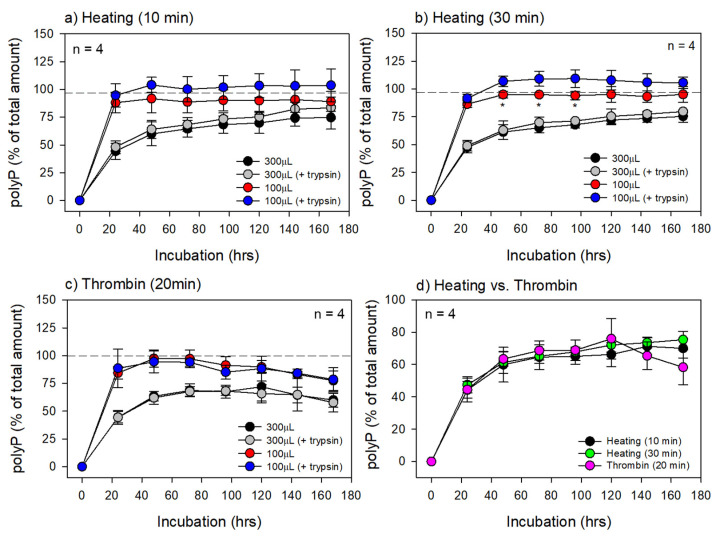
Time course changes in polyP levels in the extra-matrix PBS. Small aliquots of the PBS (10–20 μL) were collected and subjected to quantification of polyP levels. The PG matrixes were prepared by 10 min of heating (**a**) or 30 min of heating (**b**) or non-thermally by thrombin (**c**). All the data of 300 μL PG matrices and thrombin-induced gel matrices without trypsin shown in panels (**a**–**c**) are merged in panel (**d**). n = 4. * *p* < 0.05 compared with each value of trypsin treatment at the same time points.

**Table 1 biomedicines-11-02871-t001:** Water contents in individual plasma gel and fibrin gel matrices.

Treatment	Heating (10 min)		Heating (30 min)		Thrombin (20 min)	
	Weight (mg)	(Percentage)	Weight (mg)	(Percentage)	Weight (mg)	(Percentage)
Liquid	288.8 ± 9.8	100.0 ± 0.0	272.0 ± 7.1	100.0 ± 0.0	327.2 ± 2.4	100.0 ± 0.0
Gel	276.0 ± 9.0	95.6 ± 1.7	254.6 ± 8.8	88.4 ± 2.5	327.2 ± 2.4	100.0 ± 0.0
Dried	225.2 ± 7.1	78.0 ± 2.3	226.6 ± 14.2	78.6 ± 3.3	150.8 ± 10.3	46.1 ± 2.9
Water content in PG or FG	52.8 in 300 (μL)		29.4 in 300 (μL)		177.9 in 330 (μL)	

## Data Availability

Data are available from the corresponding author upon request.

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
