# Peer review of "Plasma Gel Made of Platelet-Poor Plasma: In Vitro Verification as a Carrier of Polyphosphate"

_biomedicines, 2023, doi:10.3390/biomedicines11112871_

Round 1

Reviewer 1 Report

This report by Nakamura et al. describes the potential use of a plasma-based gel isolated from pooled fractions of four healthy patients.  Gelation properties such as water content, gelation time/temperature, degradation, microstructure analysis, and polyphosphate (polyP) release were determined.  This reviewer does not feel that the methodologies used in this paper were thoroughly conducted to support the conclusions and demonstrate the usefulness of this biomaterial as a potential clinical tool.

Major concerns :

1)      There is no characterization of the components of this plasma gel. As this is a patient-derived platform, there is expected to be great variability between future samples. Without these in-depth characterization, the results presented in this manuscript is not broadly applicable. The authors suggest a component is fibrin ‘based on its fibrous appearance’ (Line 212), but no physicochemical characterization was conducted to prove this.  This lack of characterization on the composition on these gels also limits the validity of the conclusions in the Discussion: 4.3 Immunogenicity section. In Line 247 : “In general, heat denaturation is thought to modify the number of immunogenic epitopes depending on the individual proteins [19-21]. As a result, certain proteins can raise their antigen levels, while others can decrease them.”  If it is unknown what is in these gels, the effect on increase vs decrease in immunogenicity is also unknown. Their use on ‘facial esthetic treatment’ on the skin surface would be very different than if this is implanted within the body and exposed to the circulatory / lymphatic systems.

2)     Given the half life in human body is only 1.5 hrs, how does heating to 65-85C denature PolyP?  Similarly, if this needs to be heated 75C for at least 10 mins, how applicable will this be for in vivo transplantation? This would also damage nearby healthy tissue.

3)     While DAPI is a known method to detect polyphosphates, the use of DAPI in the context of this work is unclear.  Is DAPI able to distinguish ‘active’ vs ‘inactive’/degraded PolyP?  What is polymeric length cut-off for the bioactivity of PolyP?

4)     Physical characterization of this material is also very superficial and not thoroughly conducted. For example, (i) Fig 4 shows most of the polyP leaks out of the smaller 100 uL gels in the first 20-40 hrs, “However, it cannot be ruled out that the PG matrix was degraded at submicron levels to allow polyP to leak. In support of this possibility, the addition of trypsin degraded the PG matrix, as evidenced by the absence of a significant increase in the polyP release”.  This vague sentence could be better demonstrated by performing SEM after 20 hrs to show microscopic degradation.   (ii) Line 227 : “ Such hydrolytic degradation was not detected macroscopically” It is common to perform swelling experiments to determine this more quantitatively.

5)     The explanation why Trypsin degradation doesn’t increase polyP release is also unclear.

Other minor comments :

6)     For determining water content in gels, how do the authors know that the final ‘cooling and further drying in a dessicator for 60m” is enough to dry the gel completely? Has it been confirmed to dry until a constant weight has been achieved?

7)     Line 79 “ A-formulation of acid-citrate-dextrose solution (ACD-A) (Terumo, Tokyo, Japan)” – Should be “Anticoagulant Citrate Dextrose Solution”

Author Response

This report by Nakamura et al. describes the potential use of a plasma-based gel isolated from pooled fractions of four healthy patients.  Gelation properties such as water content, gelation time/temperature, degradation, microstructure analysis, and polyphosphate (polyP) release were determined.  This reviewer does not feel that the methodologies used in this paper were thoroughly conducted to support the conclusions and demonstrate the usefulness of this biomaterial as a potential clinical tool.

Response: Thank you for your consideration. We tried to address your comments as well as we could. Please consider our responses below. We hope that you will be satisfied with them.

Major concerns :

1)      There is no characterization of the components of this plasma gel. As this is a patient-derived platform, there is expected to be great variability between future samples. Without these in-depth characterization, the results presented in this manuscript is not broadly applicable. The authors suggest a component is fibrin ‘based on its fibrous appearance’ (Line 212), but no physicochemical characterization was conducted to prove this.  This lack of characterization on the composition on these gels also limits the validity of the conclusions in the Discussion: 4.3 Immunogenicity section. In Line 247 : “In general, heat denaturation is thought to modify the number of immunogenic epitopes depending on the individual proteins [19-21]. As a result, certain proteins can raise their antigen levels, while others can decrease them.”  If it is unknown what is in these gels, the effect on increase vs decrease in immunogenicity is also unknown. Their use on ‘facial esthetic treatment’ on the skin surface would be very different than if this is implanted within the body and exposed to the circulatory / lymphatic systems.

Response: Thank you for this comment. Other reviewers suggested us expand the discussion to draw the readers’ interest and curiosity regardless of our own data. To meet such requests, we to some extent expanded the discussion in several parts. However, if it seems too much for us, we did not expand it.

In our experience, we have often faced a similar dilemma in experiments using human-derived biomaterials. If we want to test their safety, we usually use immune-deficient animals, such as nude mice. However, those animals lack T cells and therefore they cannot exclude implanted materials even though these materials can be immunologically recognized as foreign matter. What we can check are the levels of toxicity and acute inflammation.

To overcome it, we have to perform a clinical study like phase I clinical trial. However, we wonder if it is the right time to perform such a clinical trial without preclinical data.

2)     Given the half life in human body is only 1.5 hrs, how does heating to 65-85C denature PolyP?  Similarly, if this needs to be heated 75C for at least 10 mins, how applicable will this be for in vivo transplantation? This would also damage nearby healthy tissue.

Response: Thank you for this comment. In the human body, it is thought that polyP is degraded mainly by phosphatases and hydrolysis. However, due to its simple chain-like structure of inorganic phosphate, polyP is not efficiently degraded or denatured in PBS by heating. Even though some phosphates are released from the polymer, we confirmed that chemical stability is well maintained and can be quantified at similar levels after 30-min heating by the DAPI-based fluorescence method.

As described in the previous publications, the heated gel is cooled in the cooling instrument or by leaving it at room temperature before clinical use. In this manuscript, this process is described in Subsection 2.3. Further investigation should be performed by biochemistry experts to address your comment.

3)     While DAPI is a known method to detect polyphosphates, the use of DAPI in the context of this work is unclear.  Is DAPI able to distinguish ‘active’ vs ‘inactive’/degraded PolyP?  What is polymeric length cut-off for the bioactivity of PolyP?

Response: Thank you for this comment. Although polyP has several forms, its major form is a linear structure. To our knowledge, there is no convincing data showing how long polyP chains can be quantified by the DAPI-based method. However, at least, it is clear (and we confirmed) that either ATP or ADP cannot sensitively be quantified by this method.

Please see Table 2 in the reference [Ushiki et al., IJMS, 2021, 22(14), 7257].

https://www.mdpi.com/1422-0067/22/14/7257

Regarding ‘active’ and ‘inactive’ PolyP, as you think, we also think that the bioactivity depends on the length. However, in human platelets, polyP length is approximately 60 on average. Thus, we speculate that this number of phosphate residues may be required for efficiently inducing cellular responses.

4)     Physical characterization of this material is also very superficial and not thoroughly conducted. For example, (i) Fig 4 shows most of the polyP leaks out of the smaller 100 uL gels in the first 20-40 hrs, “However, it cannot be ruled out that the PG matrix was degraded at submicron levels to allow polyP to leak. In support of this possibility, the addition of trypsin degraded the PG matrix, as evidenced by the absence of a significant increase in the polyP release”.  This vague sentence could be better demonstrated by performing SEM after 20 hrs to show microscopic degradation.   (ii) Line 227 : “ Such hydrolytic degradation was not detected macroscopically” It is common to perform swelling experiments to determine this more quantitatively.

Response: Thank you for this suggestion. We got a similar idea and attempted to demonstrate the possible degradation at submicron levels. However, as you may have experienced in your SEM study, we found it is almost impossible to eliminate gel debris without damaging the gel surface structure. In addition, convincing data can be obtained by comparing the same matrix before and after hydrolysis because the change could be in submicron order. Probably, we have to set up the experimental systems suitable for this purpose.

5)     The explanation why Trypsin degradation doesn’t increase polyP release is also unclear.

Response: Thank you for this comment. As described in Subsection 2.4, trypsin was added to facilitate the degradation of the PG matrix. Without any proteases, the PG matrix can be degraded by hydrolysis. However, added trypsin was expected to digest several proteins. To simulate in vivo conditions in more detail, the immune system should be included. However, the purpose of this introductory study was to characterize the basic properties of this material. Thus, we thought that the simple experimental condition would be better.

Regarding polyP release, as described in Subsections 4.2. and 4.6., this study was designed based on the possible binding of polyP to several types of protein.

Other minor comments :

6)     For determining water content in gels, how do the authors know that the final ‘cooling and further drying in a dessicator for 60m” is enough to dry the gel completely? Has it been confirmed to dry until a constant weight has been achieved?

Response: Thank you for this comment. In a preliminary study, we confirmed the time period required to reach the steady state and adopted a minimal period of time for efficient experimental performance. Macroscopically, when condensation formed on the side wall of the microtube disappeared, we stepped ahead to the next process.

7)     Line 79 “ A-formulation of acid-citrate-dextrose solution (ACD-A) (Terumo, Tokyo, Japan)” – Should be “Anticoagulant Citrate Dextrose Solution”

Response: Thank you for this suggestion. In our early studies, we were a little confused with this abbreviation. However, as described in Wikipedia, the same mixture has been called in different ways. According to the description in Wikipedia, acid-citrate-dextrose or acid-citrate-dextrose solution, also known as anticoagulant-citrate-dextrose or anticoagulant-citrate-dextrose solution (and often styled without the hyphens between the coordinate terms, thus acid citrate dextrose or ACD) is any solution of citric acid, sodium citrate, and dextrose in water. It is mainly used as an anticoagulant (in yellow top tubes) to preserve blood specimens required for tissue typing.

In the USA, “anticoagulant” seems more popular, whereas “acid” is always used in our country. Thus, we have consistently used the term “acid-citrate-dextrose” in our previous studies. Please see our related publications listed in Pubmed site below:

https://pubmed.ncbi.nlm.nih.gov/?term=kawase+tomoyuki+platelet+rich+plasma+ACD&sort=date

Reviewer 2 Report

I think this system seems interesting, but it is a little difficult to understand the gel novelty compared to other gels because of poor sentences. Today, there are many reports on gel systems of natural-derived biomaterials, such as collagen (https://doi.org/10.1016/j.addr.2003.08.004), gelatin (https://doi.org/10.3390/molecules26226795), or fibrin (https://doi.org/10.1016/j.actbio.2023.03.026). The authors should introduce these overviews and discuss the novelty of this study by comparing these papers.

1.

How about the stiffness of the gels? Did you optimize it?

2.

How about the degradation of gels? This property is also important to affect the results.

3.

There is no discussion about the comparison with other representative materials, so the reviewer cannot evaluate the system correctly.

not so bad

Author Response

I think this system seems interesting, but it is a little difficult to understand the gel novelty compared to other gels because of poor sentences. Today, there are many reports on gel systems of natural-derived biomaterials, such as collagen (https://doi.org/10.1016/j.addr.2003.08.004), gelatin (https://doi.org/10.3390/molecules26226795), or fibrin (https://doi.org/10.1016/j.actbio.2023.03.026). The authors should introduce these overviews and discuss the novelty of this study by comparing these papers.

Response: Thank you for this suggestion. We think that it is a good idea to improve this study. However, we have not compared the PG matrix with other representative gel matrices in the same experimental system. Thus, we thought that it was less convincing to compare our own data with published data. In addition, according to the suggestions by reviewer 2, we have expanded the Discussion section by adding two subsections. Thus, we are afraid that further expansion may distort the focus of this study. Please check the revised manuscript and judge the necessity of this expansion again.

1. How about the stiffness of the gels? Did you optimize it?

Response: Thank you for this comment. We did not have a rheometer or similar instruments to evaluate material stiffness. However, in previous reports [Mourão et al., Int J Growth Factors Stem Cells Dent 2018;1:64-9; Fujioka-Kobayashi et al., Platelets 2021;32:74-81], the stiffness was optimized for smooth injection. The optimal preparation conditions were 10-min of heating at 75 °C.

In a preliminary study, we confirmed that the plasma gel (PG) prepared using this protocol maintained its fluidity and enabled it to be injected through thin needles. However, when heated for 5 minutes, the resulting PG was too soft to dissolve in PBS within several minutes. Therefore, it is not suitable as a carrier. In contrast, when heated for 30 minutes, PG became brittle. Therefore, it is unsuitable for injection. However, they can still be used as carriers.

Based on this data, we prepared an experimental design and obtained the findings presented in this study.

2. How about the degradation of gels? This property is also important to affect the results.

Response: Thank you for this comment. As shown in Figure 2, the degradation started at the interface between the PG surface and PBS. As described in the first paragraph of the Discussion section, the degradation speed is largely influenced by the surface-area-to-volume ratio, although the PG contains micropores. Thus, to clearly contrast this difference, we did not use an “independent” cubic form PG, in which all the surfaces are surrounded by PBS; instead, we adopted a wall-attached form of PG, in which only the interface contacts PBS. In this comparison, we thought that we could examine the effects of the surface area-to-volume ratio on degradation.

3. There is no discussion about the comparison with other representative materials, so the reviewer cannot evaluate the system correctly.

Response: Thank you for this comment. I understand what you have meant. However, to discuss the suitability or superiority of PG over other representative materials, we believe that PG should be directly compared with those materials in the same experimental systems. As you mentioned, we think that such a comparison is important and should be performed sooner or later.

However, in this initial basic study, we aimed to introduce this easy-to-use material and demonstrate its possible application as a drug carrier. Thus, we did not think that it was appropriate to discuss these materials solely by referring to the literature.

Reviewer 3 Report

1. It is recommended to add a vertical axis label in Figure 4(b) to accurately convey the data presented in the chart. Additionally, it is suggested to adjust the horizontal axis in Figure 4(c) to fully display all data points.

2. To provide readers with a comprehensive understanding of the entire chart, it is suggested to add a description of Figure 4(d) in the caption of Figure 4.

3. It is suggested to provide the full name “Per-Arnt-Sim kinase domain” or briefly explain the meaning of “PASK domain” when it is first mentioned. Release mechanism can be strengthened by citing 10.1016/j.reactfunctpolym.2020.104501

4. According to the article, it was found that different heating times affect fiber density under electron microscopy, which may impact the mechanical performance of PG. Why not conduct direct testing of the mechanical properties of PG?

5. The result and discussion sections of the article lack a comparison between the gels formed by the heating and chemical cross-linking methods, which does not clearly highlight the advantages of the heating method. Instead, thrombin-induced gel matrices in Figure 4 appears to have a stronger retention effect on polyP.

6. The discussion section of the article does not provide further explanation of the data collected in the result section, such as water content and structural analysis under electron microscopy. Therefore, adding explanations for these parts can help readers better understand the research.

Author Response

1. It is recommended to add a vertical axis label in Figure 4(b) to accurately convey the data presented in the chart. Additionally, it is suggested to adjust the horizontal axis in Figure 4(c) to fully display all data points.

Response: Thank you for this suggestion. We modified both Figure 4(b) and (c).

2. To provide readers with a comprehensive understanding of the entire chart, it is suggested to add a description of Figure 4(d) in the caption of Figure 4.

Response: Thank you for this suggestion. In Figure 4(d), to visualize possible differences, we merged all the data of 300 μL PG matrices (without trypsin) shown in Figure 4(a), (b), and (c). This description was added to the Figure legend.

3. It is suggested to provide the full name “Per-Arnt-Sim kinase domain” or briefly explain the meaning of “PASK domain” when it is first mentioned. Release mechanism can be strengthened by citing 10.1016/j.reactfunctpolym.2020.104501

Response: Thank you for this suggestion. We replaced the abbreviation with the full name in the text.

To the best of our knowledge, your recommended article (Tong et al.) did not mention fibrin or fibrinolysis. Thus, although it is helpful to understand the general aspects of the release mechanism, we found that citation of this paper and reinforcement of the corresponding part would be difficult, at least for us, in a limited space.

Instead, we have cited several articles regarding the regulation of fibrinolysis and inserted an additional explanation in Subsection 4.3. We hope that you are satisfied with our choices.

4. According to the article, it was found that different heating times affect fiber density under electron microscopy, which may impact the mechanical performance of PG. Why not conduct direct testing of the mechanical properties of PG?

Response: Thank you for this suggestion. The reason why we did not test the mechanical properties of PG was the lack of an appropriate instrument in our lab. However, more directly, we observed changes in the physical properties. For example, overheating (>30 min) makes PG matrices brittle and unsuitable for injection. In contrast, underheating (< 5 min) increased their degradability in PBS and disabled their capacity as drug carriers. Thus, we learned about the difficulties of testing their mechanical properties.

5. The result and discussion sections of the article lack a comparison between the gels formed by the heating and chemical cross-linking methods, which does not clearly highlight the advantages of the heating method. Instead, thrombin-induced gel matrices in Figure 4 appears to have a stronger retention effect on polyP.

Response: Thank you for this comment. We observed no significant difference between the thermally- and thrombin-prepared PG matrices in Figure 4(d). However, against our working hypothesis, thrombin-prepared matrices expressed similar capacity in polyP release. We do not have either our own data or published data to explain this capacity.

As for other chemicals for the cross-linking of fibrin fibers, we added factor XIIIa, glutaraldehyde, and genipin to the Discussion. Considering each advantage and disadvantage comprehensively, we think that the thermal method has the best clinical applicability among various PG preparation methods.

In Subsection 4.4, we added an explanation of the advantages of the thermal preparation method.

6. The discussion section of the article does not provide further explanation of the data collected in the result section, such as water content and structural analysis under electron microscopy. Therefore, adding explanations for these parts can help readers better understand the research.

Response: Thank you for this comment. We added a discussion about water content in the PG matrices in Subsection 4.2.

Round 2

Reviewer 1 Report

Please see my comments to Author Response in Red below : 

1)            Response: Thank you for this comment. Other reviewers suggested us expand the discussion to draw the readers’ interest and curiosity regardless of our own data. To meet such requests, we to some extent expanded the discussion in several parts. However, if it seems too much for us, we did not expand it.

In our experience, we have often faced a similar dilemma in experiments using human-derived biomaterials. If we want to test their safety, we usually use immune-deficient animals, such as nude mice. However, those animals lack T cells and therefore they cannot exclude implanted materials even though these materials can be immunologically recognized as foreign matter. What we can check are the levels of toxicity and acute inflammation.

To overcome it, we have to perform a clinical study like phase I clinical trial. However, we wonder if it is the right time to perform such a clinical trial without preclinical data

Thank you for the responses from the authors. This reviewer agrees with the difficulties in studying the immune response to biomaterials and that this is beyond the scope of this current paper. However, my  main point here is that a detailed molecular characterization of the components that make up these gels is important. The effects on unknown immunogenicity was an example of the ambiguity about not knowing gel composition. 

For example, the authors state in Section 2.6 Line 133 that there is a “collagen matrix within the PG matrices”. Collagen is well-known a temperature-dependent gelation protein. To ensure the broad applicability of these gels, it is important to know the amount of this protein, and any other components/proteins that will affect gelation.

2) Response: Thank you for this comment. In the human body, it is thought that polyP is degraded mainly by phosphatases and hydrolysis. However, due to its simple chain-like structure of inorganic phosphate, polyP is not efficiently degraded or denatured in PBS by heating. Even though some phosphates are released from the polymer, we confirmed that chemical stability is well maintained and can be quantified at similar levels after 30-min heating by the DAPI-based fluorescence method.

As described in the previous publications, the heated gel is cooled in the cooling instrument or by leaving it at room temperature before clinical use. In this manuscript, this process is described in Subsection 2.3. Further investigation should be performed by biochemistry experts to address your comment

Please include statements about the hydrolytic stability of PolyP into discussion or introduction.

“we confirmed that chemical stability is well maintained and can be quantified at similar levels after 30-min heating by the DAPI-based fluorescence method.” This would be useful to include into the Results.

3) Response: Thank you for this comment. Although polyP has several forms, its major form is a linear structure. To our knowledge, there is no convincing data showing how long polyP chains can be quantified by the DAPI-based method. However, at least, it is clear (and we confirmed) that either ATP or ADP cannot sensitively be quantified by this method.

Please see Table 2 in the reference [Ushiki et al., IJMS, 2021, 22(14), 7257].

https://www.mdpi.com/1422-0067/22/14/7257

Limitations of the DAPI method should be included in the Discussion, such as the inability to distinguish different lengths of polyP P-subunits. As reviewed by Christ et al [Analytical Chem 2020,92, 4167], DAPI detects higher than 15 P-subunits.   

 Regarding ‘active’ and ‘inactive’ PolyP, as you think, we also think that the bioactivity depends on the length. However, in human platelets, polyP length is approximately 60 on average. Thus, we speculate that this number of phosphate residues may be required for efficiently inducing cellular responses

The PolyP quantification method used in this study does not distinguish that PolyP lengths that are released from the gel retain 60 subunits in length.

4) Response: Thank you for this suggestion. We got a similar idea and attempted to demonstrate the possible degradation at submicron levels. However, as you may have experienced in your SEM study, we found it is almost impossible to eliminate gel debris without damaging the gel surface structure. In addition, convincing data can be obtained by comparing the same matrix before and after hydrolysis because the change could be in submicron order. Probably, we have to set up the experimental systems suitable for this purpose

If performing this experiment of SEM before vs after hydrolysis will provide convincing data, then it should be performed. Otherwise, it is unclear whether the burst release in the first 24 hrs is caused by other non-degradation mechanisms such as poor gel-PolyP interactions (while maintaining PG matrix structure).

5) Response: Thank you for this comment. As described in Subsection 2.4, trypsin was added to facilitate the degradation of the PG matrix. Without any proteases, the PG matrix can be degraded by hydrolysis. However, added trypsin was expected to digest several proteins. To simulate in vivo conditions in more detail, the immune system should be included. However, the purpose of this introductory study was to characterize the basic properties of this material. Thus, we thought that the simple experimental condition would be better.

Regarding polyP release, as described in Subsections 4.2. and 4.6., this study was designed based on the possible binding of polyP to several types of protein

The explanations on the relevance of trypsin on polyP release in the manuscript are still unclear. However, the statement in the response above “designed based on the possible binding of polyP to several types of protein”  is more clear and should be added into the discussion and/or results.

Other comment : Line 256 : “Compared to the PG matrices (78−79 vol%), the thrombin gel matrix contained a greater total amount of insoluble components (approximately 46 vol%). This indicates that the water contents in the PG matrices and thrombin gel matrix were approximately 54 vol% and 21−22 vol%, respectively. This is thought to be simply due to the lack of heating during the preparation of the thrombin gel matrix. Despite this substantial difference, it remains to be elucidated why the polyP retention and release capacities were very similar between the two types of gel matrices.”

Is this reversed? Shouldn’t the PG matrices have 21-22% and Thrombin have 54% water?

Author Response

Firstly, we want to sincerely thank the reviewer for their insightful critique and thorough assessment of our work. We appreciate the time and effort they took to provide us with constructive feedback, which has been helpful in identifying areas for improvement.

Please see my comments to Author Response in Red below : 

1)            Response: Thank you for this comment. Other reviewers suggested us expand the discussion to draw the readers’ interest and curiosity regardless of our own data. To meet such requests, we to some extent expanded the discussion in several parts. However, if it seems too much for us, we did not expand it.

In our experience, we have often faced a similar dilemma in experiments using human-derived biomaterials. If we want to test their safety, we usually use immune-deficient animals, such as nude mice. However, those animals lack T cells and therefore they cannot exclude implanted materials even though these materials can be immunologically recognized as foreign matter. What we can check are the levels of toxicity and acute inflammation.

To overcome it, we have to perform a clinical study like phase I clinical trial. However, we wonder if it is the right time to perform such a clinical trial without preclinical data

Thank you for the responses from the authors. This reviewer agrees with the difficulties in studying the immune response to biomaterials and that this is beyond the scope of this current paper. However, my main point here is that a detailed molecular characterization of the components that make up these gels is important. The effects on unknown immunogenicity was an example of the ambiguity about not knowing gel composition. 

For example, the authors state in Section 2.6 Line 133 that there is a “collagen matrix within the PG matrices”. Collagen is well-known a temperature-dependent gelation protein. To ensure the broad applicability of these gels, it is important to know the amount of this protein, and any other components/proteins that will affect gelation.

Response 2: Thank you for this suggestion. As described in the Results section in the original version, judging from the SEM observations, we believe that the major protein component is fibrinogen/fibrin. However, although it is not insolubilized or gelated by itself, owing to its major population, we cannot exclude the possibility that serum albumin could be insolubilized along with the conversion from fibrinogen to fibrin. However, no studies have indicated the major components of the PG matrix. To accurately quantify the amounts of these components and other minor components in detail, chemical analytical methods should be adopted, and the purpose of this study should be changed.

It should be noted again that the major advantages of this material are the use of autologous (patients’ own) blood and a simple preparation protocol at point-of-care. The purpose of this study was to verify the applicability of the PG matrix for the retention and release of polyP in a time-dependent manner. Similar to PRP and PRF, a high cost-benefit ratio was also highlighted when verified.

Unfortunately, we have no instruments or skills to perform detailed molecular characterization of the components that make up these gels, as no one has performed such a characterization for PRP or PRF. Thus, we have added a statement in the Discussion section (the end of subsection 4.1.) in which further chemical analysis is required to clarify the detailed molecular contents of the PG matrix.

2) Response: Thank you for this comment. In the human body, it is thought that polyP is degraded mainly by phosphatases and hydrolysis. However, due to its simple chain-like structure of inorganic phosphate, polyP is not efficiently degraded or denatured in PBS by heating. Even though some phosphates are released from the polymer, we confirmed that chemical stability is well maintained and can be quantified at similar levels after 30-min heating by the DAPI-based fluorescence method.

As described in the previous publications, the heated gel is cooled in the cooling instrument or by leaving it at room temperature before clinical use. In this manuscript, this process is described in Subsection 2.3. Further investigation should be performed by biochemistry experts to address your comment

Please include statements about the hydrolytic stability of PolyP into discussion or introduction.

“we confirmed that chemical stability is well maintained and can be quantified at similar levels after 30-min heating by the DAPI-based fluorescence method.” This would be useful to include into the Results.

Response 2: Thank you for this comment. To avoid hydrolytic degradation, longer chains (~600 Pi) of polyP must be stored at -80 ℃ by the manufacturer. Storing a longer polyP at 4 °C is inappropriate and degrades it. Regarding the middle length (~60-Pi) polyP, we confirmed its chemical stability using the DAPI-based method. The manufacturer recommends storage in a refrigerator. To our knowledge, middle-length polyP can be stored at 4 °C without significantly reducing the DAPI-dependent signal. On the other hand, heating polyP in liquid PBS at 75 °C for 30 min did not reduce DAPI-dependent signals. Thus, we have added this explanation to the Results section (Fig. 4).

However, to add this information to the Introduction, at least a number of publications have been published. To add this to the Discussion section, we need our own data obtained from appropriate biochemical analyses. Thus, adding this explanation to the Introduction or Discussion may not be appropriate.

3) While DAPI is a known method to detect polyphosphates, the use of DAPI in the context of this work is unclear.  Is DAPI able to distinguish ‘active’ vs ‘inactive’/degraded PolyP?  What is polymeric length cut-off for the bioactivity of PolyP?

Response: Thank you for this comment. Although polyP has several forms, its major form is a linear structure. To our knowledge, there is no convincing data showing how long polyP chains can be quantified by the DAPI-based method. However, at least, it is clear (and we confirmed) that either ATP or ADP cannot sensitively be quantified by this method.

Please see Table 2 in the reference [Ushiki et al., IJMS, 2021, 22(14), 7257].

https://www.mdpi.com/1422-0067/22/14/7257

Limitations of the DAPI method should be included in the Discussion, such as the inability to distinguish different lengths of polyP P-subunits. As reviewed by Christ et al [Analytical Chem 2020,92, 4167], DAPI detects higher than 15 P-subunits.   

Regarding ‘active’ and ‘inactive’ PolyP, as you think, we also think that the bioactivity depends on the length. However, in human platelets, polyP length is approximately 60 on average. Thus, we speculate that this number of phosphate residues may be required for efficiently inducing cellular responses

The PolyP quantification method used in this study does not distinguish that PolyP lengths that are released from the gel retain 60 subunits in length.

 Response 2: Thank you for this comment. This is a significant limitation of the DAPI-based method. However, this method cannot monitor the length of polyP. However, this does not indicate that all the polyP molecules released from the PG gel are shorter than 60 Pi. Thus, this limitation does not severely influence data interpretation. We have added this limitation and our thoughts in subsection 4.7.

4) Response: Thank you for this suggestion. We got a similar idea and attempted to demonstrate the possible degradation at submicron levels. However, as you may have experienced in your SEM study, we found it is almost impossible to eliminate gel debris without damaging the gel surface structure. In addition, convincing data can be obtained by comparing the same matrix before and after hydrolysis because the change could be in submicron order. Probably, we have to set up the experimental systems suitable for this purpose

If performing this experiment of SEM before vs after hydrolysis will provide convincing data, then it should be performed. Otherwise, it is unclear whether the burst release in the first 24 hrs is caused by other non-degradation mechanisms such as poor gel-PolyP interactions (while maintaining PG matrix structure).

Response 2: Thank you for this comment. We now understand what you have meant. As we replied, we confirmed in the preliminary study that, despite possible submicron-level initial degradation, it is almost only possible to detect it by examining the same area of the same subject before and after hydrolysis.

However, this does not imply that we can exclude other possibilities. In the preliminary study, we also observed that before starting incubation in PBS, the PG matrix was extensively washed, but polyP was released similarly. Thus, this finding suggests that a significant number of polyP molecules weakly attached to the surface of the PG matrix were detached in the initial phase.

Instead, polyP molecules trapped (not chemically bound) in the semi-dried fibrin mesh may be released by hydration without the hydrolysis of fibrin fibers and polyP by itself. We have added this possibility to expand the discussion.

We have added to the discussion at the beginning of subsection 4.3.

5) Response: Thank you for this comment. As described in Subsection 2.4, trypsin was added to facilitate the degradation of the PG matrix. Without any proteases, the PG matrix can be degraded by hydrolysis. However, added trypsin was expected to digest several proteins. To simulate in vivo conditions in more detail, the immune system should be included. However, the purpose of this introductory study was to characterize the basic properties of this material. Thus, we thought that the simple experimental condition would be better.

Regarding polyP release, as described in Subsections 4.2. and 4.6., this study was designed based on the possible binding of polyP to several types of protein

The explanations on the relevance of trypsin on polyP release in the manuscript are still unclear. However, the statement in the response above “designed based on the possible binding of polyP to several types of protein”  is more clear and should be added into the discussion and/or results.

Response 2: Thank you for this comment. It is very difficult to simulate the complex in vivo thrombosis and thrombolysis situation. Thus, we set up a simple “model” and used trypsin that can nonspecifically degrade various types of proteins efficiently in the absence of calcium.

Although degradation of the PG matrix was observed macroscopically, the addition of trypsin facilitated polyP release. However, this effect was not statistically significant. This finding suggests that the trypsin model may not be the best for our purpose. However, it may depend on the balance between the amount of added polyP and that of its binding proteins or its binding non-protein molecules. We added the discussion in the subsection 4.7.

Other comment : Line 256 : “Compared to the PG matrices (78−79 vol%), the thrombin gel matrix contained a greater total amount of insoluble components (approximately 46 vol%). This indicates that the water contents in the PG matrices and thrombin gel matrix were approximately 54 vol% and 21−22 vol%, respectively. This is thought to be simply due to the lack of heating during the preparation of the thrombin gel matrix. Despite this substantial difference, it remains to be elucidated why the polyP retention and release capacities were very similar between the two types of gel matrices.”

Is this reversed? Shouldn’t the PG matrices have 21-22% and Thrombin have 54% water?

Response 2: Thank you for this question. It is a fact that the PG matrix contained fewer amounts of water than the thrombin gel matrix. The dissimilarity in outcomes is probably a direct consequence of the heating procedure, which induces the depletion of water content via the process of evaporation.

Round 3

Reviewer 1 Report

Thank you to the authors for making these revisions and providing more details about the limitations of the methods used in this study.  There are 2 more minor comments :

1) Overall the quality of the English language in this paper is well written. However, some of the recent additions/revisions require improvement. 

2) For the last previous Reviewer comment : "This indicates that the water contents in the PG matrices and thrombin gel matrix were approximately 54 vol% and 21-22 vol%, respectively."

Is this reversed? Shouldn’t the PG matrices have 21-22% and Thrombin have 54% water?"

Response 2: Thank you for this question. It is a fact that the PG matrix contained fewer amounts of water than the thrombin gel matrix. The dissimilarity in outcomes is probably a direct consequence of the heating procedure, which induces the depletion of water content via the process of evaporation.

Reviewer comment 2 : For this, I meant were the water content values for PG (54%) and  thrombin (21-22%) mixed up in the text?  As the author also states, PG has lower water content than thrombin, but in the text it is written as PG having more water.

Overall the quality of the English language in this paper is well written. However, some of the recent additions/revisions require improvement. 

Author Response

Thank you to the authors for making these revisions and providing more details about the limitations of the methods used in this study.  There are 2 more minor comments :

1) Overall the quality of the English language in this paper is well written. However, some of the recent additions/revisions require improvement.

2) For the last previous Reviewer comment : "This indicates that the water contents in the PG matrices and thrombin gel matrix were approximately 54 vol% and 21-22 vol%, respectively."

Is this reversed? Shouldn’t the PG matrices have 21-22% and Thrombin have 54% water?"

Response 2: Thank you for this question. It is a fact that the PG matrix contained fewer amounts of water than the thrombin gel matrix. The dissimilarity in outcomes is probably a direct consequence of the heating procedure, which induces the depletion of water content via the process of evaporation.

Reviewer comment 2 : For this, I meant were the water content values for PG (54%) and  thrombin (21-22%) mixed up in the text?  As the author also states, PG has lower water content than thrombin, but in the text it is written as PG having more water.

Response 3: Thank you for pointing this out. Upon re-evaluation of our data and text, you are correct. We made an error in our presentation of the values. The correct water content for the PG matrix should indeed be 21-22% while the thrombin gel matrix has approximately 54% water content. We apologize for the oversight and will amend this immediately in the manuscript to reflect accurate data.

Before the original submission, we had made our English expression checked by Editage, a professional editing service. However, the revised version may include minor errors. We hope that minor English language mistakes will be corrected during the MDPI editing process.
